# TARP-VP: Towards Evaluation of Transferred Adversarial Robustness and Privacy on Label Mapping Visual Prompting Models

**Zhen Chen**[1]    **Yi Zhang**[2]    **Fu Wang**[3]    **Xingyu Zhao**[2]    **Xiaowei Huang**[1]    **Wenjie Ruan**[4]*

[1]University of Liverpool    [2]University of Warwick    [3]University of Exeter    [4]USTC

{cz97, xiaowei.huang}@liverpool.ac.uk
{yi.zhang.16, xingyu.zhao}@warwick.ac.uk
fw377@exeter.ac.uk   rwjie@ustc.edu.cn

## Abstract

Adversarial robustness and privacy of deep learning (DL) models are two widely studied topics in AI security. Adversarial training (AT) is an effective approach to improve the robustness of DL models against adversarial attacks. However, while models with AT demonstrate enhanced robustness, they become more susceptible to membership inference attacks (MIAs), thus increasing the risk of privacy leakage. This indicates a negative trade-off between adversarial robustness and privacy in general deep learning models. Visual prompting is a novel model reprogramming (MR) technique used for fine-tuning pre-trained models, achieving good performance in vision tasks, especially when combined with the label mapping technique. However, the performance of label-mapping-based visual prompting (LM-VP) under adversarial attacks and MIAs lacks evaluation. In this work, we regard the MR of LM-VP as a unified entity, referred to as the LM-VP model, and take a step toward jointly evaluating the adversarial robustness and privacy of LM-VP models. Experimental results show that the choice of pre-trained models significantly affects the white-box adversarial robustness of LM-VP, and standard AT even substantially degrades its performance. In contrast, transfer AT-trained LM-VP achieves a good trade-off between transferred adversarial robustness and privacy, a finding that has been consistently validated across various pre-trained models.

## 1   Introduction

Deep learning models have gained great success, yet concerns regarding their security continue to grow, as they are susceptible to various attacks [30, 13, 23, 29, 35]. In addition to the deep learning models, training samples are also the key to this success. Consequently, attacks targeting the relationship between training samples and models have emerged, such as adversarial attacks [23, 14, 43, 37] and membership inference attacks (MIAs) [25, 28, 30, 32]. Adversarial attacks are gradient-based methods that introduce imperceptible perturbations on inputs, generating adversarial examples (AEs) that cause the target models to give incorrect predictions. Adversarial training (AT) [23], proposed by Szegedy *et al.* has been recognized as one of the most effective defenses against such attacks. The basic idea of AT is to incorporate AEs into the training process, resulting in significantly improved performance under adversarial attacks compared to standard training (ST).

MIAs, on the other hand, aim to determine whether a specific sample was part of the model's training data. Various ideas exist for performing MIAs, *e.g.*, shadow model-based attacks [30] and prediction

---

*Corresponding Author

38th Conference on Neural Information Processing Systems (NeurIPS 2024).

confidence-based attacks [39, 28, 32]. The success of these attacks largely depends on the model's generalization error during training and testing [31]. Therefore, models with lower generalization errors are inherently more resistant to such attacks.

AT-trained models exhibit more severe privacy risks compared to ST: (1) larger generalization error, manifested in both natural and adversarial examples [32]; (2) higher sensitivity on training data compared with ST [32]; (3) robust overfitting [27], where the model's adversarial robustness declines despite the natural accuracy continuing to increase at a certain training stage. These issues result in a negative trade-off between adversarial robustness and privacy. As illustrated in Fig.1, while AT significantly enhances adversarial robustness, it is more susceptible to MIAs, especially after robust overfitting, *i.e.*, between 100-150 epochs in Fig.1d, where the MIA success rate increases significantly. **Notably, the above conclusions are only for general deep-learning models.**

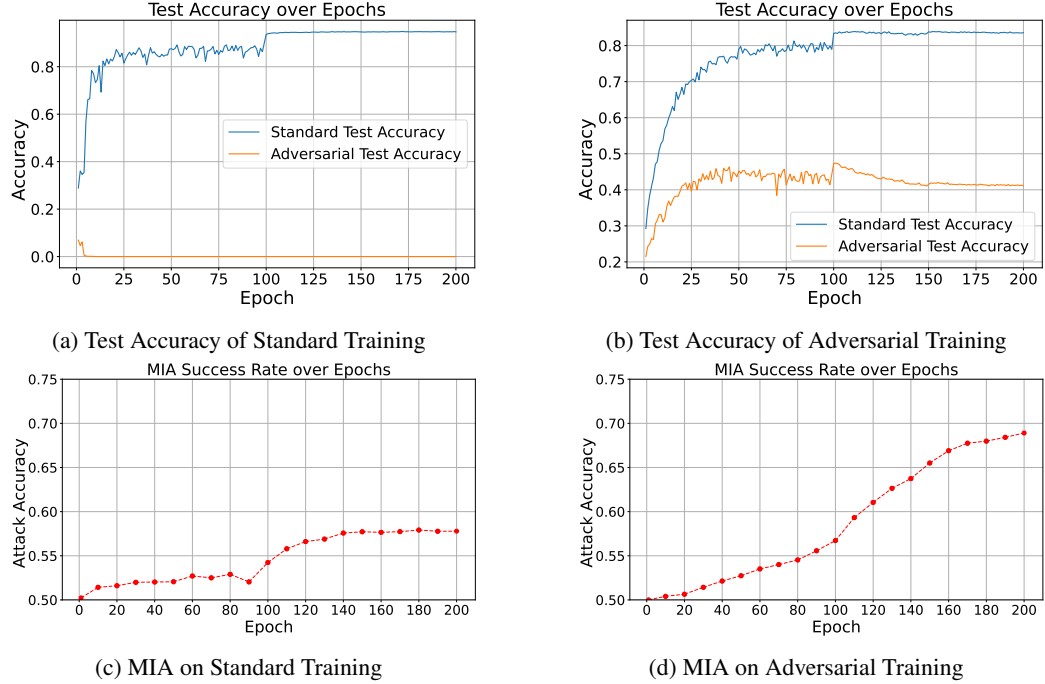

(a) Test Accuracy of Standard Training

(b) Test Accuracy of Adversarial Training

(c) MIA on Standard Training

(d) MIA on Adversarial Training

Figure 1: Trade-off between test accuracy and membership inference attacks of standard training and adversarial training along with training on CIFAR-10 with $\ell_\infty$ threat model using ResNet18.

Visual prompting (VP) [3] is a model reprogramming (MR) [10, 7] technique for pre-trained models, used for downstream image classification tasks. Initially, VP involves adding a single, input-agnostic prompt to input images to enhance a pre-trained model's generalization ability. Label mapping (LP) further improves VP's performance by mapping source labels to target labels, denoted as LM-VP, which exhibits strong performance in downstream tasks [3]. In this paper, we regard a general pre-trained model after LM-VP as a new model, and its security remains under-explored, including its susceptibility against AEs and MIAs, and compatibility with AT. We consider two forms of AT: the standard AT for white-box adversarial robustness and transfer AT for black-box transferred adversarial robustness, *i.e.*, generating adversarial examples through another threat model. We empirically demonstrate that the intuitions and relationships between adversarial robustness and privacy observed in general models do not always hold for the LM-VP model.

In summary, our contributions lie in:

- From a novel perspective of considering LM-VP as a distinct model, we conduct the first evaluation of its security, *i.e.*, (transferred) adversarial robustness and privacy;

- Based on the concept of transfer attacks, we implement transferred adversarial training for the LM-VP model to enhance its transferred adversarial robustness;

- We empirically demonstrate that intuitions regarding privacy in general models do not necessarily apply to LM-VP models. Furthermore, we show that standard AT is invalid

for LM-VP, while transfer AT on LM-VP exhibits a superior trade-off between transferred adversarial robustness and privacy across various pre-trained models.

## 2 Related Work and Background

### 2.1 Visual Prompting

The prompt technique [18, 20, 15] was originally employed in NLP tasks. In essence, it modifies the original input text to enhance specific task performance without altering the parameters of the pre-trained model. For instance, prompts indicating sentiment can be added for text classification [24], while prompts indicating the target language can be used for translation tasks [42]. Bahang *et al.*[3] first transfers this idea to computer vision tasks, *i.e.*, VP. Compared to traditional transfer learning methods like fine-tuning [33] and linear probes [1], VP does not modify the parameters of the pre-trained model. Instead, it alters the original image by adding prompts, *i.e.*, introducing additional pixels, enabling task-specific and input-agnostic adjustments. During VP training, only the prompts and output transformation are updated. VP exhibits strong performance across various datasets and significantly reduces the training parameters compared to traditional transfer learning methods. Output transformation, or label mapping (LM), is another key technique contributing to VP's performance. Chen *et al.*[6] proposes the iterative label mapping (ILM) method to replace the random mapping in vanilla VP. Arif *et al.*[2] and Li *et al.*[19] apply a trainable fully-connected layer for label mapping, achieving promising results and improved efficiency. Li *et al.*[19] also explore the VP in training differential private models using the PATE framework [26], while Chen *et al.*[5] investigate VP in test-time adversarial robustness by implementing adversarial prompts during testing.

### 2.2 Adversarial Robustness Evaluation

Adversarial robustness is an important metric for evaluating model robustness, referring to a model's performance under adversarial attacks. Adversarial attacks target image gradients, iteratively introducing imperceptible perturbations to images to generate AEs. These AEs often cause standard-trained deep learning models to misclassify them with high confidence. Commonly used adversarial attacks include PGD [23], FGSM [14], and CW [4] attacks. AT is an approach to improve the adversarial robustness of deep learning models. It can be formulated as a min-max problem, where the inner maximization searches for perturbations that maximize the loss, while the outer minimization optimizes the model, *i.e.*,

$$\min_{\boldsymbol{\theta}} \mathbb{E}_{(\boldsymbol{Z},y)\sim\mathcal{D}} \left[ \max_{\|\boldsymbol{\delta}\|\leq\epsilon} L\left(f_{\boldsymbol{\theta}}(\boldsymbol{X}+\boldsymbol{\delta}), y\right) \right]. \tag{1}$$

Various strategies exist to solve the inner maximization in AT, including PGD-AT [23], TRADES [43], and MART [37], etc. LOAT[40] boosts AT via a Fisher-Rao norm-based regularization, SEAT[36] extends AT to medical segmentation and FAAL[44] ensure both robustness and fairness during AT, Chen *et al.*[8] proposes NRAT to enhance adversarial robustness under noisy labels. While robustness evaluation mainly focuses on discriminative models, Zhang *et al.*[45] introduces a robustness notion of text-to-image (T2I) generative models and proposes the ProTIP framework for evaluation.

### 2.3 Membership Inference Attacks

MIAs refer to determining whether a given data point was part of the training data for a trained model, raising significant data privacy concerns. Shokri *et al.*[30] introduce the first MIA on classification models, *i.e.*, shadow training attack, which involves creating several shadow models that simulate the target model, with these shadow models trained on data records similar to those used for training the target model. An attack model is then trained to recognize the relationship between the members of the shadow models' training data and the shadow models' outputs, which turns out to be a binary classification. This attack model can subsequently infer the membership of the target model's training dataset. There are two findings in this work [30]: (1) the higher the degree of the model's overfitting, the higher the attack's success rate, and (2) the more complex the training dataset, the higher the MIA success rate. Intuitively, increasing the number of shadow models improves attack performance by providing more samples for training the attack model, but this also requires more computational resources. Yeom *et al.*[39] propose threshold-based MIAs, which compare the target model's prediction confidence for the true label against a certain threshold. This method achieves

performance similar to the shadow model training method with significantly reduced computational resource consumption. Song *et al.*[31] further proposes the class-dependent thresholds for a more powerful attack and implements the MIA based on prediction entropy.

## 3 Robustness and Privacy Evaluation of Visual Prompting Models

In this section, we first provide an overall design of LM-VP models, including the prompt designs, label mapping techniques, LM-VP model training, as well as the tricks we adopt. Then, we analyze the white-box adversarial robustness of LM-VP, demonstrating that pre-trained models largely influence its white-box adversarial robustness. We further propose transfer AT to enhance black-box transferred adversarial robustness. Finally, we analyze the intuition between LM-VP models and privacy.

### 3.1 Label Mapping Visual Prompting

LM-VP model aims to keep the parameters of the pre-trained models fixed while performing input transformation and output transformations, *i.e.*, label mapping. Therefore, we divide the LM-VP model into three parts: (1) prompt generation; (2) label mapping; and (3) model training.

#### 3.1.1 Prompt Generation

For the prompt generation, we first rescale the images from the target domain under a certain rescale ratio to a size smaller than that of the source domain. Then, trainable noise $w_1$, *i.e.*, prompt is added around the image to ensure the final image size matches that of the source domain. Therefore, the length (height and width) of each pixel patch $p$ is

$$p = 1/2[(H_1 - H_2) + (W_1 - W_2)], \tag{2}$$

where $H_1$ and $W_1$ represent the height and width of the source domain, $H_2$ and $W_2$ represent the height and width of the rescaled target domain, then the final shape of prompts $P$ is

$$P = C \times [H_1/p + W_1/p - 4] \times p^2, \tag{3}$$

where $C$ is the image channels, $[H_1/p + W_1/p - 4]$ represents the amount of pixel patches in each channel. In vanilla VP [3], they rescale the target images to the size of the source domain and replace the edges with random noise, *i.e.*, prompts. In comparison, although the final prompts $P$ are the same, we preserve the edge information of the target domain images, which is more intuitively reasonable as we retain all the information of target images. Preserving the edge information might help increase the correlation between prompts and images during training. Fig. 2 shows the difference between these two ways of prompt generation.

#### 3.1.2 Label Mapping

For output transformation, vanilla VP uses random mapping, randomly selecting some labels from the source domain to match those of the target domains and discarding the remaining unused labels. However, this random mapping often leads to a performance drop in VP as it may ignore some important information in the unused labels [19]. Therefore, we consider label mapping as a trainable component: using a fully connected layer to train the mapping from source labels to target labels, which is similar to [19] and [2]. Consequently, in LM-VP, we introduce two trainable parameters: the prompt noise parameters $w_1$ and the parameters $w_2$ in the label mapping layer $f_\ell(w_2; \theta_2)$.

#### 3.1.3 LM-VP Model Training

For LM-VP model training, our objective is to modify the prompts by updating the noise parameters $w_1$, and the parameters $w_2$ of FC layers of the label mapping. During the testing phase, the same optimized prompts are applied to all test data, fed into the frozen pre-trained model, and finally output the label mapping results. LM-VP relies on the pre-trained model and label mapping, indicating that it does not require a large target dataset. Therefore, we can train LM-VP models using a subset of the target dataset without a significant performance drop compared with training on the entire target dataset, which significantly improves training efficiency. In contrast, general models tend to underfitting when trained on small subsets, leading to poor generalization performance. Although the LM-VP model does not heavily rely on training data, insufficient data can still affect its generalization.

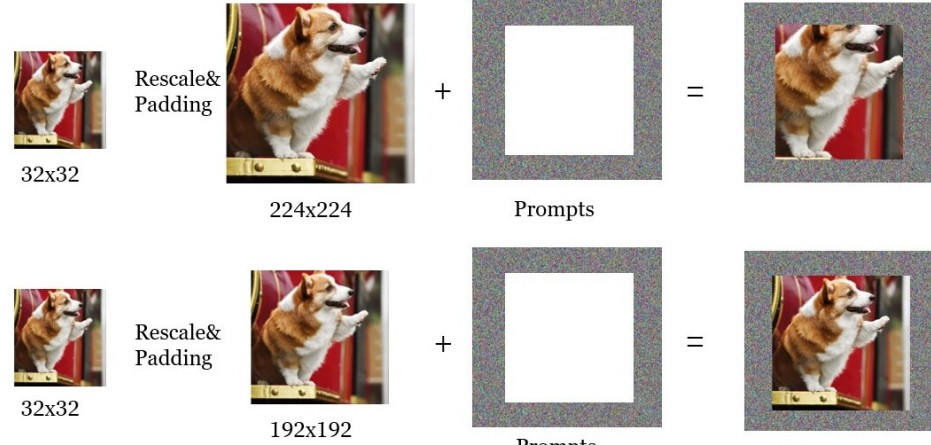

Figure 2: Two ways to add prompts: (1) Top: rescale a target image to the source domain size and replace the edge of the image with prompts; (2) Bottom: rescale a target image to a size smaller than the source domain and add prompts to make it the same size as source domain.

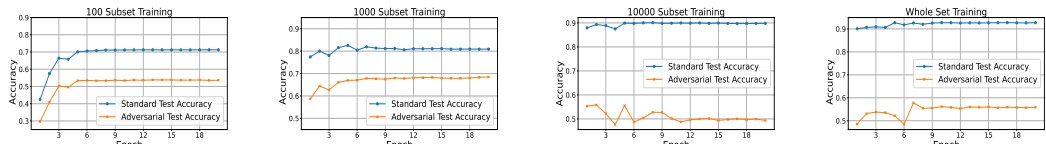

Figure 3: LM-VP model (pre-trained on Swin Transformer[21]) performance on the whole test set using standard training with different numbers of training subsets (random 100, 1000, 10000 subsets and whole training set) on CIFAR-10, transferred adversarial robustness is evaluated on $\ell_\infty$ threat model using ResNet18.

Therefore, we use the SAM [12] version of SGD optimizer to update the weights of $w_1$ to improve the LM-VP model's generalization, *i.e.*,

$$w_{t+1} = w_t - \eta \left( \nabla L \left( w_t + \epsilon_t \right) + \lambda w \right), \tag{4}$$

where $\eta$ is the learning rate, $\epsilon$ is the parameter to maximize the loss function $L$, and $\lambda$ is the weight decay. Given the prediction on source domain $\hat{y}_S$, the final prediction label on target domain $\hat{y}_T$ is obtained by

$$\hat{y}_T = \text{softmax} \left( f_\ell \left( \omega_2; \hat{y}_S \right) \right). \tag{5}$$

Fig. 3 illustrates the impact of data volume on VP performance during training, demonstrating that insufficient training data may not always hurt the performance of the LM-VP model, *e.g.*, the LM-VP model trained with a subset of 1000 samples (second figure) achieves the best transferred adversarial robustness; the LM-VP model trained with a subset of 10000 samples (third figure) achieves a similar standard test accuracy compared with the LM-VP model trained with the whole training set. Additionally, subset training significantly reduces the running time to 0.12x, 0.25x, and 0.45x on subsets of 100, 1,000, and 10,000 samples respectively.

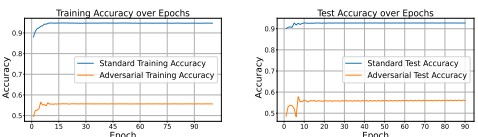

Figure 4: Training and testing performance of LM-VP (pre-trained on Swin Transformer) using standard training on CIFAR-10, transferred adversarial robustness is evaluated on $\ell_\infty$ threat model using ResNet18.

The LM-VP model also exhibits the property of rapid convergence, *i.e.*, it can quickly achieve a near-optimal performance and then remain steady with continued training, for both the natural samples and adversarial samples, as shown in Fig. 4, it takes about 10 epochs to convergence for both

training and testing. Therefore we can set a small number of epochs to improve training efficiency on LM-VP models.

## 3.2 White-box Adversarial Robustness of LM-VP Models

A crucial distinction between the LM-VP model and a general model lies in the presence of a pre-trained model that does not participate in training[3]. Using white-box adversarial robustness metrics to evaluate LM-VP models can be heavily influenced by the choice of pre-trained models, as shown in Table. 1 (Standard Training), there is no clear pattern in their best adversarial robustness across different pre-

Table 1: Best performance(%) on CIFAR-10 with different pre-trained models in Standard-Trained LM-VP models and Standard AT-Trained LM-VP models under white-box adversarial attacks.

| Pre-trained models | Standard Training | | Adversarial Training | |
|---|---|---|---|---|
| | Natural$_{te}$ | PGD-20 | Natural$_{te}$ | PGD-20 |
| **ResNet50** | 80.52 | 8.33 | 23.10 | 0.8 |
| **ResNet152** | 84.76 | 57.09 | 14.24 | 0 |
| **Wideresnet** | 80.91 | 40.29 | 12.15 | 0 |
| **VIT** | 91.50 | 19.28 | 27.78 | 0 |
| **Swin** | 92.00 | 0 | 34.65 | 0 |
| **ConvNext** | 97.97 | 43.22 | 40.69 | 0 |

trained models, thus VP may play a limited role in defending against the white-box adversarial attack. From Fig. 5, for standard-trained LM-VP models, the best (highest) adversarial robustness was only observed in the early stages of training, as training progresses, for all pre-trained models, the adversarial robustness continues to decline until it reaches a stabilized status, thus the best adversarial robustness may largely reflect the pre-trained models' inherent adversarial robustness when transferred to the target dataset.

We also notice that the standard AT is invalid in LM-VP, with really poor results in both natural and adversarial performance, see Table. 1 (Adversarial Training). Since the LM-VP model is trained on the target dataset, but the generation of adversarial examples depends on a fixed pre-trained model from the source dataset domain, the domain shift may lead to unsatisfactory results of AT on the target dataset.

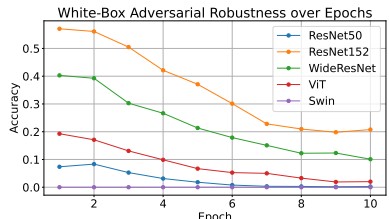

Figure 5: Epoch-wise white-box adversarial robustness of LM-VP using standard training on CIFAR-10.

Regarding how pre-trained models affect downstream adversarial robustness, [38]and [34] provide more insights, *e.g.*, Yamada *et al.*[38] conclude that network architecture is a strong source of robustness in transfer learning. In this sense, different pre-trained models may lead to different boundary relationships between adversarial robustness and privacy, evaluating LM-VP using white-box adversarial attacks may make it difficult to reach consistent conclusions.

## 3.3 LM-VP models with Transferred Adversarial Training

For the evaluation of the transferred adversarial robustness of the LM-VP model, we use another general model as the threat model to produce adversarial examples and train the LM-VP model to defend against them. In this scenario, the intensity of the transfer attack remains constant once the threat model is selected. This consistency holds true regardless of the chosen pre-trained model. This inherent consistency is thus helpful for exploring and establishing a sensible relationship between transferred adversarial robustness and privacy within LM-VP models. Compared to white-box adversarial robustness which is heavily influenced by the pre-trained models, utilizing transferred adversarial robustness serves as a more reliable and insightful evaluation method for LM-VP models.

Within the framework of transfer AT, the LM-VP model which comprises VP, a pre-trained model, and LM, is treated as a unified black-box system, A fixed-parameter threat model, excluded from the training process, is employed to generate AEs $x'$, and then train LM-VP models using adversarial loss, this transfer AT consistently optimizes in the same direction since the attack remains constant.

### 3.4 Relationship between LM-VP models and Training Dataset Privacy

In our work, we use the resistance of a model against MIAs to reflect its privacy. According to [32] and [31], AT-trained general models exhibit larger generalization errors and higher sensitivity to training data, wherein generalization errors and sensitivity are two key factors that influence the success of MIAs on general models, thus indicating a contradiction between adversarial robustness and privacy for general models. However, in the LM-VP model, the architecture of pre-trained models appears to be the more significant factor affecting its (white-box) standard adversarial robustness (Table. 1). As a result, we are not able to establish a reliable boundary relationship between its standard adversarial robustness and privacy, which has prompted us to focus primarily on the relationship between its transferred adversarial robustness and privacy.

In the training of LM-VP models, the parameters (weights and biases) of the pre-trained models are fixed, and the trainable parameters are noise parameters and label mapping parameters. This significantly reduces the influence of the input (both natural and adversarial samples) on the LM-VP model compared to general models, which also enables effective training of LM-VP models with a small subset of data, indicating lower sensitivity of LM-VP models to training data, the results in Fig. 3 also support this statement. Additionally, the generalization ability of LM-VP models mainly relies on the pre-trained models, which have been trained on large-scale datasets and learned rich universal features. With fixed model parameters, the risk of overfitting is reduced.

As shown in Fig. 4, during LM-VP standard training and transfer AT, both training accuracy and test accuracy on natural examples and adversarial examples are very close. Based on this empirical evidence, LM-VP models exhibit minimal generalization error and low sensitivity to training data, which should intuitively enhance their resistance to MIAs and better protect the privacy of training data. However, apart from generalization errors and sensitivity, there might be other factors influencing the LM-VP model's resistance to MIAs, such as the prior knowledge embedded in different pre-trained models. In our experiments, we empirically demonstrate that MIA analyses applied to general models do not always hold for LM-VP models, and transferred adversarial robustness and privacy can be improved simultaneously using transfer AT.

## 4 Experiments

In this section, we conduct comprehensive experiments to evaluate the performance of LM-VP models under transferred adversarial attacks and a threshold-based MIA. Regarding the trade-off among standard accuracy, transferred adversarial robustness, and MIA success rate, we comprehensively compare different pre-trained models in LM-VP models. We conduct main experiments[2] on CIFAR-10 and additional experiments on Tiny-ImageNet to show the good generalization performance of transfer AT. We implement all experiments on a server with an RTX3090 GPU.

### 4.1 Experimental Setup

For adversarial attacks, all experiments follow the standard settings: $\ell_\infty$ threat models for all methods, the perturbation limit $\epsilon = 8/255$, and step size 2/255, we mainly choose ResNet18 as the threat model. For LM-VP models, we use the source models pre-trained on 224x224 ImageNet. For training, we follow the settings in [19], *i.e.*, SGD with SAM technique and a momentum of 0.9 to optimize the LM-VP defense models, the total training epoch is 20. We choose ResNet50 [17], ResNet152 [17], WideResNet-50-2 [41], VIT [9], Swin Transformer [21], ConvNext [22], and EVA [11] models to show the effect of different pre-trained source models in LM-VP model training.

For evaluation, (1) adversarial attacks: we choose PGD-20 and CW-20; (2) MIA: we implement the threshold-based attack on both the natural examples and adversarial examples.

### 4.2 Classification Evaluation of Standard Trained LM-VP Models

To evaluate the performance of LM-VP models using standard training, *i.e.*, its training loss is given by

$$\ell^{ST}\left(\mathbf{x}_i, y_i, \boldsymbol{\theta}\right) = \mathrm{CE}\left(f_{\boldsymbol{\theta}}\left(\mathbf{x}_i + P\right), y_i\right). \tag{6}$$

---

[2]Code is available at `https://github.com/TrustAI/TARP-VP`

Table 2: Best performance(%) on CIFAR-10 with different pre-trained models in Standard-Trained LM-VP models under Threat models ResNet18 or WRN-34-10.

| Best Performance on natural examples and adversarial examples | | | | | | | |
|---|---|---|---|---|---|---|---|
| Pre-trained models | Threat models | Natural$_{tr}$ | Natural$_{te}$ | PGD-10$_{tr}$ | PGD-20 | CW-20 | T/E |
| **ResNet50** | | 87.73 | 86.30 | 31.14 | 35.61 | 34.30 | 251s |
| **ResNet152** | | 90.39 | 89.51 | 36.76 | 35.99 | 35.67 | 440s |
| **WRN-50-2** | | 87.77 | 86.78 | 37.73 | 39.76 | 38.90 | 381s |
| **VIT** | ResNet18 | 94.91 | 92.67 | 51.25 | 51.95 | 50.70 | 589s |
| **Swin** | | 94.78 | 92.71 | 56.46 | 57.80 | 57.34 | 1025s |
| **ConvNext** | | 99.33 | 98.28 | **88.70** | **89.11** | **89.37** | 2116s |
| **EVA** | | **99.66** | **98.54** | 86.95 | 87.40 | 87.56 | 2674s |
| **Best Performance on natural examples and adversarial examples** | | | | | | | |
| Pre-trained models | Threat models | Natural$_{tr}$ | Natural$_{te}$ | PGD-10$_{tr}$ | PGD-20 | CW-20 | T/E |
| **ResNet50** | | 87.18 | 85.87 | 30.33 | 32.32 | 30.98 | - |
| **ResNet152** | | 89.95 | 89.42 | 37.24 | 37.26 | 37.08 | - |
| **WRN-50-2** | | 87.97 | 87.01 | 38.25 | 41.36 | 39.90 | - |
| **VIT** | WRN-34-10 | 94.78 | 92.77 | 51.41 | 52.23 | 52.12 | - |
| **Swin** | | 95.08 | 92.8 | 55.23 | 59.20 | 57.54 | - |
| **ConvNext** | | 99.19 | 98.03 | **88.20** | **88.51** | **88.23** | - |
| **EVA** | | **99.64** | **98.45** | 86.21 | 86.98 | 87.24 | - |

$P$ is the prompt. We report its natural accuracy and transferred adversarial robustness in Table. 2, where the "**best performance**" refers to the performance under the epoch of the best (standard or transferred) adversarial robustness.

Based on the parameter capacity of the pre-trained models, we regard them as small, medium, and large models. Specifically, ResNet50, ResNet152, and WRN-50-2 are small models, ViT and Swin Transformer are medium models, and ConvNext and EVA are large models. We report only the natural accuracy during training, while different attacks are employed during testing. The results in Table. 2 show that: (1) There is a clear hierarchy in natural accuracy and transferred adversarial robustness based on the size of the pre-trained models, *i.e.*, small models have a natural accuracy below 90%, medium models around 92%, and large models around 98%; for transferred adversarial robustness, small models are below 45%, medium models range from 50% to 60%, and large models range from 85% to 90%; (2) In LM-VP models, the transferred adversarial robustness is not significantly affected by the size of the threat model, *i.e.*, a larger threat model (WRN-34-10) may not be more challenge compared with ResNet18. For instance, small and medium models sometimes have higher transferred adversarial robustness achieved under WRN-34-10; (3) In LM-VP models, even when more attack steps are used, the transferred adversarial robustness of the test data often remains higher than that of the training data set, although more attack steps being considered more powerful (comparing PGD-10$_{tr}$ and PGD-20).

### 4.3 Classification Evaluation of Transferred AT-based LM-VP Models

We implement the transfer AT of LM-VP models proposed in Section 3.3 and report the performance in Table. 3. Specifically, during training, we use PGD attack with 10 steps to generate AEs:

$$x^0 = x + \sigma, \text{ where } \sigma \sim \mathcal{N}(0, 1), \tag{7}$$

$$x^{t+1} = \Pi_{x+\mathcal{S}}(x^t + \alpha sign(\nabla_x \mathcal{L}(\theta, x^t, y))), \tag{8}$$

$x^0$ is obtained by perturbing $x$ with random noise $\sigma$ sampled from the normal distribution $\mathcal{N}(0, 1)$, $t$ denotes the current attack step, $\alpha$ is the step size, $\Pi$ denotes the projection function, $\mathcal{S} \subseteq \mathbb{R}^d$ denotes the perturbation set of AEs, we train LM-VP models with the following training loss:

$$\ell^{AT}(\mathbf{x}_i, y_i, \boldsymbol{\theta}) = \text{CE}(f_{\boldsymbol{\theta}}(\mathbf{x}'_i + P), y_i), \tag{9}$$

where $\mathbf{x}'_i$ denotes the AE after PGD. To ensure consistency with the ST in Table. 2, we report the results under the same metrics. Consistent with our findings in Section 4.2, AEs generated by different

Table 3: Best performance(%) on CIFAR-10 with different pre-trained models in Transfered AT-Trained LM-VP models under Threat model ResNet18.

| Pre-trained models | Threat models | Natural$_{tr}$ | Natural$_{te}$ | PGD-10$_{tr}$ | PGD-20 | CW-20 | T/E |
|---|---|---|---|---|---|---|---|
| **Best Performance on natural examples and adversarial examples** | | | | | | | |
| **ResNet50** | | 68.84 | 70.37 | 64.10 | 63.01 | 61.78 | 671s |
| **ResNet152** | | 68.83 | 77.08 | 63.39 | 63.95 | 62.92 | 950s |
| **WRN-50-2** | | 69.68 | 70.42 | 62.07 | 62.86 | 60.89 | 875s |
| **VIT** | ResNet18 | 86.23 | 86.64 | 77.49 | 75.34 | 74.87 | 1380s |
| **Swin** | | 89.32 | 89.74 | 80.72 | 79.14 | 77.89 | 2205s |
| **ConvNext** | | 97.79 | 98.02 | 92.61 | 91.63 | 91.02 | 3446s |
| **EVA** | | **98.64** | **98.32** | **93.19** | **92.43** | **91.50** | 4136s |

Table 4: MIA success rate(%) on CIFAR-10 with different pre-trained models in Standard and Transferred AT Trained LM-VP models under Threat model ResNet18.

| Pre-trained models | Standard Training | | Transferred AT | |
|---|---|---|---|---|
| **Generation Gap and MIA Success Rate on Trained LM-VP Models** | | | | |
| | MIA Nat | MIA Adv | MIA Nat | MIA Adv |
| **ResNet50** | 68.92 | 57.88 | 55.27 | 51.19 |
| **ResNet152** | 75.34 | 56.46 | 62.15 | 50.77 |
| **WRN-50-2** | 62.58 | 50.66 | 50.46 | 50.94 |
| **VIT** | 51.66 | 50.37 | 50.53 | 51.78 |
| **Swin** | 51.75 | 50.53 | 50.23 | 51.63 |
| **ConvNext** | 80.14 | 77.33 | 50.32 | 50.70 |
| **EVA** | 77.46 | 73.35 | 50.32 | 50.67 |

threat models have minimal impact on both training and testing. Therefore, we exclude the results for WRN-34-10 as a threat model.

The results in Table. 3 indicate that: (1) Transferred AT significantly enhances the transferred adversarial robustness at the cost of reduced natural accuracy. Specifically, the transferred adversarial robustness improvement is around 20%-35% for small models, 20%-25% for medium models, and 3%-6% for large models; (2) Compared to the standard-trained LM-VP models, the transferred adversarial robustness of the training set (PGD-10$_{tr}$) is usually higher than that of the test set (PGD-20) in transferred AT-trained LM-VP models.

## 4.4 Privacy Evaluation of LM-VP Models

In this section, we evaluate the privacy performance of LM-VP models under the threshold-based MIA. Our attack implementation is based on [16], the MIA success rate with a threshold $\eta$ is given by:

$$MIA(\eta) = \frac{1}{2} \times \left( \frac{\sum_{(x,y) \in D_{\text{train}}} \mathbf{1}\left[ f_\theta(x)_y \geq \eta \right]}{|D_{\text{train}}|} + \frac{\sum_{(x,y) \in D_{\text{test}}} \mathbf{1}\left[ f_\theta(x)_y < \eta \right]}{|D_{\text{test}}|} \right), \quad (10)$$

where the $\eta_{optim}$ is obtained by computing all possible $\eta$ that maximizes the MIA success rate, i.e.,

$$\eta_{\text{optim}} = \arg\max_\eta MIA(\eta). \quad (11)$$

This attack is mainly based on the model generalization error, i.e., models with higher generalization error are more susceptible to the attack. Conversely, the success rate should approach 50% for a model with little generalization error. However, this principle does not always consistently apply to LM-VP models. From Table. 4, for standard-trained LM-VP models, only the VIT and Swin Transformer can effectively resist the attack with an MIA success rate near 50%. Another observation is the MIA success rate on AEs is lower for the standard-trained LM-VP models.

For transferred-AT trained LM-VP models, the MIA success rate for most cases is around 50%, markedly reducing the risk of training data privacy leakage. This indicates that transferred adversarial

Table 5: Best performance(%) on Tiny-ImageNet with different pre-trained models in Standard-Trained LM-VP models and Transfered AT-Trained LM-VP models under threat model ResNet18.

| Pre-trained models | Standard Training | | | Transfer Adversarial Training | | |
|---|---|---|---|---|---|---|
| | Natural$_{te}$ | PGD-20 | MIA Nat | Natural$_{te}$ | PGD-20 | MIA Nat |
| **ResNet50** | 62.74 | 10.26 | 57.46 | 50.42 | 34.60 | 50.90 |
| **ResNet152** | 65.00 | 20.53 | 62.14 | 57.36 | 38.81 | 50.85 |
| **WRN-50-2** | 70.12 | 16.59 | 53.50 | 50.50 | 30.59 | 50.89 |
| **VIT** | 80.97 | 37.77 | 54.00 | 72.02 | 50.22 | 51.45 |
| **Swin** | 79.93 | 41.81 | 56.95 | 75.08 | 55.81 | 51.35 |
| **ConvNext** | **89.01** | 73.47 | 58.47 | 87.60 | **76.61** | 52.04 |

robustness and privacy can be simultaneously achieved in LM-VP models. One plausible explanation is that during transfer AT, the original training examples are perturbed before feeding into the model, thus these data are not exposed to the trained model, this may help mitigate the MIA issue since LM-VP models also do not suffer from large generalization error (Table. 2 and Table. 3) and increased training data sensitivity (Fig. 3) and transfer AT do not train the original training examples.

### 4.5 Results on Tiny-ImageNet

To demonstrate the efficiency of transferred AT on LM-VP models, we provide the results on Tiny-ImageNet which has a resolution of 64x64 and contains 200 classes, results shown in Table. 5, LM-VP models with transfer AT improve transfer adversarial robustness by 3%-24% and mitigate the MIA success rate by 3%-12% compared to LM-VP models with standard training.

## 5 Conclusion

In this paper, we regard the models trained using the LM-VP technique as a novel model type and analyze its adversarial robustness and privacy. We observe that the choice of pre-trained models significantly influences the white-box adversarial robustness of LM-VP, making it hard to draw consistent conclusions. Therefore, we focus more on its transferred adversarial robustness and its interaction with MIA-based privacy. To address both concerns, we propose the transfer AT method for LM-VP models to enhance performance on both fronts. Experiments across various pre-trained models demonstrate that: *(i) both standard-trained and transfer AT-trained LM-VP models show a positive correlation between transferred adversarial robustness and pre-trained model size*, and *(ii) transfer AT significantly boosts the transferred adversarial robustness of LM-VP models while also enhancing its training data privacy*. These findings indicate the advantage of LM-VP models trained with transfer AT in AI security. In future work, we will integrate relevant theories from related domains to delve deeper into the security implications of LM-VP models.

## Acknowledgments

ZC's contribution is supported by the University of Liverpool and China Scholarship Council (CSC). XH's contribution is supported by the UK EPSRC through End-to-End Conceptual Guarding of Neural Architectures [EP/T026995/1]. XZ's contribution is supported by the UK EPSRC New Investigator Award through Harnessing Synthetic Data Fidelity for Assured Perception of Autonomous Vehicles.

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

# A    Appendix / supplemental material

## A.1    Ablation Studies on Rescale Factor

In [19], they conclude the impact of the rescale ratio on the training of LM-VP models, *i.e.*, a larger rescale ratio yields better performance. However, an excessively large rescale ratio can also lead to overfitting to the target domain. In this section, we compare two rescale ratios, corresponding to the two cases shown in Fig. 2. In terms of performance, there is only a minimal gap between the two ways.

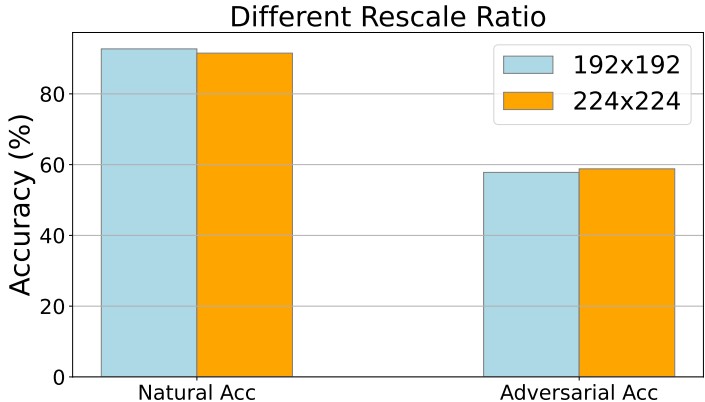

Figure 6: Comparison on different rescale ratios. The pre-trained model is Swin Transformer.

## A.2    Ablation Studies on Perturbation Limit

We set $\epsilon = 4/255$ and evaluate the LM-VP model based on the ResNet50 and ConvNext pre-trained models. The result is consistent with our main experiment where $\epsilon = 8/255$, *i.e.*, transfer AT achieves better transferred adversarial robustness and privacy trade-offs at the cost of natural accuracy.

Table 6: Best performance(%) on CIFAR-10 with two pre-trained models in Standard-Trained LM-VP models and Transfered AT-Trained LM-VP models under $\epsilon = 4/255$ of threat model ResNet18.

| Pre-trained models | Standard Training | | | Transfer Adversarial Training | | |
|---|---|---|---|---|---|---|
| | $\text{Natural}_{te}$ | PGD-20 | MIA Nat | $\text{Natural}_{te}$ | PGD-20 | MIA Nat |
| ResNet50 | 84.90 | 33.19 | 73.99 | 67.96 | 65.10 | 51.91 |
| ConvNext | 97.72 | 86.86 | 79.84 | 97.68 | 89.77 | 51.20 |

