# OpenReview forum: "TARP-VP: Towards Evaluation of Transferred  Adversarial Robustness and Privacy on Label  Mapping Visual Prompting Models"
_NeurIPS.cc/2024/Conference — NeurIPS 2024 poster_

### Official Review · Reviewer_RUkH · 2024-07-11

**Soundness:** 3
**Presentation:** 3
**Contribution:** 3
**Rating:** 7
**Confidence:** 4

**Summary:**

This paper investigates the adversarial robustness and privacy aspects of models trained using the Language Model Visual Prompting (LM-VP) technique, which has not been done before. The results suggest that LM-VP models trained with transfer AT have advantages in AI security.

**Strengths:**

- This paper connects two larger topics: privacy and prompt learning. The novelty is high, because the reviewer has not seen any comparable work before.
- The authors also contribute to transferred adversarial training, which is very important for domain adaptation and connected to label mapping.
- The researchers demonstrated that applying transferred adversarial training (AT) to Language Model-Vision Pretraining (LM-VP) models yields superior trade-offs between adversarial robustness and privacy protection. This improvement was consistently observed across a diverse range of pre-trained models examined in the study.
- The authors ablate two type of prompt generation which shows that they take a deeper look to this topic.

**Weaknesses:**

- Ln267-268: "MIA success rate near 50%"  is the statement in the discussion section. 50% means for me random. It is missing for me, if this  is an improvement or not.  I am asking myself if there a baseline exists which could be even lower.  It should have been shortly mentioned how to order this in general in MIA defenses.

Comments on writing:
 - Ln83: No AutoAttack [1] mentioned, but later in Ln219.
 - Eq1: A dot instead of a comma at the end of line.
 - Eq3: The multiplications signs look like a convolution.

[1] https://robustbench.github.io/#div_imagenet_Linf_heading
[2] https://ml.cs.tsinghua.edu.cn/ares-bench/#/leaderboard

**Questions:**

- In the experimental setup, it is not clear to the reviewer, which MIA attack did you use? In Ln102 the authors state two threshold-based MIAs. Is it one of these? Or both?

**Limitations:**

- Adversarial attack setup and limitations: An ablation study on different epsilon sizes would give more insights on the limitations. Ln212: The attack epsilon size 8/255 (for CIFAR-10) is larger than the standard size for adversarial training is 4/255 [1,2] for ImageNet.

In light of the current results, these limitations only could be interesting for the appendix as additional insights and does not have to appear in the  main sections.

---

> ### Author Rebuttal · Authors · 2024-08-07
>
> **W1**. Generally, in existing work, the MIA success rate is typically larger than 50%. A value close to 50% indicates that the attack is invalid, as this is like a random guess. Thus, a successful defense against MIA would result in an MIA success rate close to 50%, as seen in Table 4 of [1], Table 3 of [2], and Tables 2, and 3 of [3]. In these studies, some defenses reduce MIA from 60%-80% to around 50%, but not below 50%.
>
> **W2**. We appreciate for pointing out these errors in writing, we will correct them in our revised manuscript.
>
> **Q1**. The MIA we use is based on Yeom et. al [4], and actually, Song et al. [2] also evaluate MIA and use the MIA based on Yeom [4]. We grouped the two references in Ln102, which may cause ambiguity, and we will clarify this in our manuscript.
>
> **L1**. Actually, we use epsilon size 8/255 as it is a commonly used setting when using ResNet18 or WideResNet as threat models on CIFAR-10 after [5]. As per the reviewer’s suggestion, we select two pre-trained models in LM-VP models to conduct experiments under epsilon size 4/255, although the results are slightly different with 8/255, we see similar trends, and we will complete the whole ablation experiments in our manuscript.
> | Pre-trained models |ST-Natural|ST-PGD|ST-MIA|Transfer AT-Natural|Transfer AT-PGD|Transfer AT-MIA|
> |--------------------|:-------------------------------:|:----------------------------:|-----------------------------|:-------------------------------------------:|:----------------------------------------:|:---------------------------------------:|
> | ResNet50           | 84.90                         | 33.19                      | 73.99                       | 67.96                                     | 65.10                                  | 51.91                                 |
> | ConvNext           | 97.72                         | 86.86                      | 79.84                       | 97.68                                     | 89.77                                  | 51.20                                 |
>
>
> [1] Milad Nasr, Reza Shokri, and Amir Houmansadr. Machine learning with membership privacy using adversarial regularization. In Proceedings of the 2018 ACM SIGSAC conference on computer and communications security, pages 634–646, 2018.
>
> [2] Shokri R, Stronati M, Song C, et al. Membership inference attacks against machine learning models[C]//2017 IEEE symposium on security and privacy (SP). IEEE, 2017: 3-18.
>
> [3] Liwei Song, Reza Shokri, and Prateek Mittal. Privacy risks of securing machine learning models against adversarial examples. In Proceedings of the 2019 ACM SIGSAC Conference on Computer and Communications Security, pages 241–257, 2019.
>
> [4] Samuel Yeom, Irene Giacomelli, Matt Fredrikson, and Somesh Jha. Privacy risk in machine learning: Analyzing the connection to overfitting. In 2018 IEEE 31st computer security foundations symposium (CSF), pages 268–282. IEEE, 2018.
>
> [5] Hongyang Zhang, Yaodong Yu, Jiantao Jiao, Eric Xing, Laurent El Ghaoui, and Michael Jordan.Theoretically principled trade-off between robustness and accuracy. In International conference on machine learning, pages 7472–7482. PMLR, 2019

---

> > ### Comment · Reviewer_RUkH · 2024-08-13
> >
> > Thank you for your thorough response. I appreciate the time you took to address each of the points I raised.
> > Well, you could have taken 1/255 or 16/255 as well as epsilon size for the proposed ablation study.

---

> > > ### Author Response · Authors · 2024-08-14
> > >
> > > Thank you for your reply and suggestions. We will add more epsilon size options in our updated version to explore the ablation effects of this parameter.

---

### Official Review · Reviewer_bbnn · 2024-07-13

**Soundness:** 3
**Presentation:** 3
**Contribution:** 3
**Rating:** 7
**Confidence:** 3

**Summary:**

This paper explores the trade-offs between adversarial robustness and privacy in deep learning models, highlighting that while AT improves robustness but it increases vulnerability to MIA. The authors introduce an ANF-based graph structure and CryptoANFNet, a neural network model for cryptographic problem-solving, demonstrating that their approach achieves a good balance between robustness and privacy.

**Strengths:**

This paper considers both robustness and privacy issues, which are very valuable topics in the DNN training area.

**Weaknesses:**

NA

**Questions:**

No

---

> ### Author Rebuttal · Authors · 2024-08-07
>
> We appreciate the reviewers' recognition of our work. Our main contribution is that we first introduce a method that simultaneously enhances transfer adversarial robustness and privacy. As a new research prospect, we are happy to discuss any questions you may have. Additionally, we will also fully release our code to enhance the contribution of this work to the AI security community.

---

### Official Review · Reviewer_J3te · 2024-07-13

**Soundness:** 1
**Presentation:** 2
**Contribution:** 1
**Rating:** 4
**Confidence:** 5

**Summary:**

The works shows that LM-VP models can achieve the great adversarial robustness and privacy at the same time, different from full model adversarial training. Across different pre-trained models, the proposed transferred adversarial training achieves good classification accuracy and low MIA success rates.

**Strengths:**

1. Different pre-trained models are tried, across ResNet50, ViT, Swin, and ConvNext, etc.
2. The motivation is clear, AT has bad trade off between robustness and privacy while the LM-VP might be a possible solution.

**Weaknesses:**

1.  Considering the efficiency of LM-VP adaptation, why not try different datasets other than CIFAR10? The current results are restricted to CIFAR10, which is a 10-class classification problem and low resolution (which can perform OK in LM-VP setting). But what about a higher resolution dataset with more classes?
2. The main concerns come from the effectiveness of adversarial attacks against LM-VP, which also raises the concern about the high adversarial robustness mentioned in the paper. The high robust accuracy for ConvNext and EVA might be due to the low transfer attack success rates from ResNet. Can you also show me the success rates of transfer attacks generated on ResNet-18 to attack ConvNext and EVA? If this is very low, it is not surprising that so-called robust accuracy is similar to standard accuracy.
3. It is still not clear to me why standard white-box adversarial attacks can not be applied to LM-VP models. From my perspective, both LM and VP should be treated as parameters and the adversarial attacks can be applied accordingly.
4. In Section 3.3, it is stated that the trainable parameters are noise parameters. However, the LM as a fc layer has also parameters.

**Questions:**

See Weakness.

**Limitations:**

Theoretical work is considered limitation, which is claimed to be solved in the future.

---

> ### Author Rebuttal · Authors · 2024-08-07
>
> **W1**. As per the reviewer’s suggestion, we conducted experiments on Tiny-ImageNet, which has a resolution of 64x64 and contains 200 classes. We select different pre-trained models and the results indicate that: The LM-VP model with transfer AT improves transfer adversarial robustness by 3%-24% and mitigates the MIA success rate by 3%-12% compared to the LM-VP model with standard training, on both CIFAR-10 and Tiny-ImageNet, transfer AT on the LM-VP model demonstrates a better robustness-privacy trade-off, which shows a good generalization performance.
> |Pre-trained models|ST-Natural|ST-PGD|ST-MIA|Transfer AT-Natural|Transfer AT-PGD|Transfer AT-MIA|
> |:-:|:-:|:-:|:-:|:-:|:-:|:-:|
> | ResNet50|62.74| 10.26 | 57.46| 50.42|34.60| 50.90|
> | ResNet152|65.00| 20.53| 62.14| 57.36| 38.81| 50.85|
> | WRN-50-2|70.12| 16.59| 53.50| 50.50| 30.59| 50.89|
> | VIT| 80.97| 37.77| 54.00| 72.02| 50.22| 51.45|
> | Swin| 79.93| 41.81| 56.95|75.08| 55.81| 51.35|
> | ConvNext| **89.01**| 73.47| 58.47| 87.60| **76.61**| 52.04|
>
> **W2**.  We thank the reviewer for pointing out this concern. Regarding this, we guess the reviewer might think that the transfer adversarial attacks generated by ResNet-18 are weak. However, from Table 1 in our manuscript, for pre-trained models other than ConvNext and EVA, the transfer adversarial attacks pose a significant threat to them, evidenced by a substantial gap in natural accuracy and adversarial robustness. The choice of pre-trained models is important in the LM-VP models, ConvNext and EVA have 197.96M and 304.14M parameters, respectively, and are fully pre-trained on ImageNet. We have provided the results of adversarial robustness against ResNet-18 transfer attacks on the LM-VP models using ConvNext and EVA (Table 1 and Table 2 in our manuscript which indicate the attack success rate (1-adversarial robustness), high adversarial robustness shows their good synergy with LM-VP models.
>
> We speculate the reviewer wants to see the ResNet-18 transfer adversarial attacks on ConvNext and EVA when transferred to CIFAR-10 after traditional finetune. After a full finetune, the standard accuracy reaches nearly 100% and the **attack success rate is 12.50% for ConvNext and 18.75% for EVA, with MIA success rates of 62.1% and 76.80% respectively**, finetune results are similar to LM-VP results, indicating that ConvNext and EVA are indeed robust to such transfer attacks on CIFAR-10 after transfer learning, but they are vulnerable to MIA.
>
> Lastly, we want to emphasize that our work does not solely consider the transfer adversarial robustness as the only metric. If we only consider this, it is evidence that it is highly dependent on the performance of the pre-trained model (Table 1 in our manuscript ), e.g., large-size pre-trained models achieve better transfer adversarial robustness. However, we also address privacy concerns associated with the LM-VP model, ConvNext and EVA achieve high adversarial robustness, but they still have much room for improvement in terms of privacy (Table 3 in our manuscript: Standard trained LM-VP models). Applying transfer AT to the LM-VP models not only further enhances adversarial robustness but also mitigates privacy issues, which is the primary contribution we claim.
>
> **W3**. We thank the reviewer for raising this question and rethinking the white-box attack on LM-VP models. According to our new attempt, white-box adversarial attacks LM-VP models can be indeed applied to LM-VP models. However, as shown in the table below, the adversarial robustness of the LM-VP model varies greatly when selecting different pre-trained models, making it difficult to obtain a consistent conclusion. Different from a general model, the pre-trained model plays an important role in the LM-VP model but the parameters are fixed and do not participate in training, VP plays limit role in defending against the white-box adversarial attack, thus its adversarial robustness may largely reflect the pre-trained models’ inherent adversarial robustness when transferred to the target dataset. Furthermore, we study the standard form of AT in the LM-VP model which is different from the transfer AT in our manuscript.
> |Pre-trained models|ST-Natural|ST-PGD|AT-Natural|AT-PGD|
> |:-:|:-:|:-:|:-:|:-:|
> |ResNet50|80.52|8.33|23.10|0.8|
> |ResNet152|84.76|57.09|14.24|0|
> |Wideresnet| 80.91| 40.29|12.15|0|
> |VIT|91.50|19.28|27.78|0|
> |Swin|92.00|0|34.65|0|
> |ConvNext|97.97|43.22|40.69|0|
>
> Unlike transfer AT, which can improve transfer adversarial robustness, standard AT can be regarded as invalid in the LM-VP model. It has really poor performance in both natural accuracy and standard adversarial robustness. The reasons may lie in:
>
> (1) The adversarial examples generated by LM-VP models rely heavily on pre-trained models. The information of the source dataset may lead to unsatisfactory results of training with those adversarial examples on the target dataset.
>
> (2) The adversarial perturbation directly affects VP, which further damages VP performance during AT.
>
> We choose different pre-trained models to illustrate the above situation regarding the white-box adversarial robustness. However, our work focuses on studying the transfer adversarial robustness and privacy of the LM-VP model and proves the effectiveness of transfer AT. The same transfer attack model (ResNet18) is used to ensure the consistency of the experimental environment. Considering the white-box adversarial attack on the LM-VP model, its essence may be to attack fixed pre-trained models. How to study or improve its adversarial robustness is an interesting topic. We will revise the corresponding part in our manuscript and add the analysis and experiments of the LM-VP model under white-box adversarial attacks.
>
> **W4**. We thank the reviewer for pointing out this mistake, LM as an FC layer, also has parameters, like in Section 3.1.3 we mentioned the parameters w2 of the fc layer, this is a writing error in Section 3.3, and we will correct it.

---

> ### Comment · Reviewer_J3te · 2024-08-08
>
> I still think that the so-called transferred adversarial robustness is far away from the adversarial robustness.
>
> 1. In the paper, "adversarial robustness" is used multiple times, Line 156, Line 254, Line 272. However, it is "transferred adversarial robustness" according to the author's response.
> 2. In the second table of your rebuttal, ResNet50 standard test accuracy is 80.52% and in Table 1 of your submission, ResNet50 standard test accuracy is 86.3%. I am confused about your setup.
> 3. In the second table of your rebuttal, ResNet152 standard PGD accuracy is 57.09% and is even higher than 35.99% in your submission, as transfer attack PGD20 accuracy. I am confused why the white-box attacks are even worse.
> 4. I need authors to double check the adversarial attack setting in both the original submission and the rebuttal.

---

> ### Author Response · Authors · 2024-08-09
>
> We agree that transferred adversarial robustness is not exactly the same as adversarial robustness; to clarify this, we will use the term "transferred adversarial robustness" to make our work as precise as possible in the updated version, we appreciate the reviewer pointing this out. We also agree that the white-box standard adversarial robustness of LM-VP models is a topic worth studying. However, we want to provide some concerns about why evaluating the standard adversarial robustness of LM-VP models can be challenging. In contrast, the black-box transfer adversarial robustness of LM-VP models does not suffer from the same issues described below.
>
> - **The standard adversarial robustness of LM-VP models may largely reflect more about the inherent properties of a fixed pretrained model rather than an effective evaluation of the LM and VP components**. In Table 2 of our rebuttal, there are significant differences in the best adversarial robustness of LM-VP models with different pretrained models. Despite using the same VP and LM components, their best adversarial robustness varies greatly and lacks a clear pattern, e.g., larger size pretrained models may not have better (or worse) adversarial robustness. This indicates the standard adversarial robustness may significantly be influenced by inherent features of the fixed pretrained model rather than the whole LM-VP model, whereas our goal is to evaluate the whole LM-VP model rather than just the pretrained component.
> -  **Whether the standard adversarial robustness makes sense for LM-VP models or not**: Adversarial examples generated by the LM-VP model are significantly influenced by the fixed pretrained model trained on the **source dataset**. The validity of using these samples to assess the LM-VP model's adversarial robustness on the **target dataset** requires further consideration.
>
> **Q1**. Since our manuscript focuses on the transferred adversarial robustness of LM-VP models, we frequently use the terms “transfer” or “transferred” throughout the paper, e.g., we mention we enhanced the transferred adversarial robustness in the contribution in Ln 57. The original manuscript does not consider the standard adversarial robustness, thus in some parts, we use adversarial robustness as transferred adversarial robustness. During rebuttal, since we rethink the standard adversarial robustness in LM-VP, we will check and correct the relevant statements in the revised manuscript and add the analysis of the standard adversarial robustness.
>
> **Q2**. The two columns on the left of Table 2 in the rebuttal are the results after standard training on LM-VP models. **“Best performance” refers to the performance under the epoch of the best (standard or transferred) adversarial robustness**, i.e., in Table 2 of rebuttal, when the standard PGD-20 adversarial robustness is best at 8.33%, the corresponding natural accuracy is 80.52% under that epoch; In contrast, in the manuscript, when the ResNet18 transfer PGD-20 adversarial robustness is best at 35.61%, it has a natural accuracy of 86.3%. We will add the description of the term “best performance” in the revised manuscript to clarify this, as it can indeed be easily misunderstood.
>
> **Q3**. Table 2 of the rebuttal reflects the standard adversarial robustness of LM-VP models. Based on these two values alone (57.09% and 35.99%), we can only infer that, at a specific stage of LM-VP model training, the adversarial examples generated by ResNet18 are more aggressive than those generated by the LM-VP model based on pretrained ResNet152. For the general trained model, we generally believe that white-box attacks are stronger than black-box transfer attacks. However, the LM-VP model is influenced by pretrained models, which are trained on source datasets. It is conceivable that the adversarial examples under some pretrained models are strong and some are weak. For example, from Table 2 of the rebuttal, pretrained Swin shows an extremely strong white-box adversarial attack while pretrained ResNet152 is relatively weak. However, the strength of ResNet18 transfer adversarial attack remains consistent.
>
> We would also like to mention that 57.09% of the standard adversarial robustness occurs only in the first epoch of training. This early-stage standard adversarial robustness likely reflects the inherent properties of the pretrained ResNet152, but its standard adversarial robustness decreases along with training, e.g., only 19.81% remains at epoch 9. Conversely, the ResNet18 transfer adversarial robustness maintains a relatively stable level, fluctuating between 32% and 36% for pretrained ResNet152.
>
> **Q4**. Regarding the ResNet18, WRN-34-10 transfer adversarial attack, and the standard adversarial attack, our settings are consistent: PGD with epsilon=8/255, num_steps=10, step_size=2/255 for training, and only change the num_steps=20 for testing, the above settings are the same in the manuscript and rebuttal.

---

> > ### Comment · Reviewer_J3te · 2024-08-09
> >
> > First, thank you for your patient rebuttal and response. If I understand correctly:
> > 1. You are studying transferred adversarial robustness for LM-VP model trained with transferred AT;
> > 2. You did not study adversarial robustness by the white-box attacks because it is not consistent when evaluating LM-VP with different pretrained models due to Table 2 first two columns;
> >
> > Then, I want to finalize my opinion to both you and **AC**:
> > 1. the adversarial robustness should be the worst case (the strongest) attack when evaluating a model, which means, at least the stronger one between the transferred attack you have tried by PGD against ResNet18 and the PGD you tried during the rebuttal.
> > 2. If you find that the PGD-20 against the victim model is even weaker than PGD-20 against another substitute model (ResNet-18 or WRN-34-10), PGD is not a good way to evaluate adversarial robustness in your experimental setup. Therefore, your so-called transferred adversarial robustness can not reflect the adversarial robustness. Then your so-called the better trade-off between adversarial robustness and privacy might be not convincing.
> > 3. The main assumption by the paper is wrong: "Notably, the LM-VP models are incapable of generating standard forms of AEs like general models due to the input transformation"; "LM-VP models do not inherently have the traditional adversarial robustness property". This has been admitted by the authors in the rebuttal as well by their PGD-20 experiments on Swin. You can argue with **AC** about this.

---

> > > ### Author Response · Authors · 2024-08-12
> > >
> > > We once again thank the reviewer for the valuable feedback.
> > >
> > > In this response, we aim to further clarify the experimental results we provided in our previous response regarding PGD-20 attacks. Additionally, as we clarified in our response, this paper does not intend to claim that transfer adversarial attacks should replace white-box attacks for evaluating adversarial robustness. Rather, the main purpose of this paper is to systematically evaluate both standard training and transfer AT across various pre-trained models on LM-VPs, **focusing on the trade-off between transferred adversarial robustness and privacy**. Below are our responses:
> > >
> > > 1. Yes, we study transfer adversarial robustness and its relationship with MIA-based privacy for LM-VP models.
> > >
> > > 2. We did not study the white-box adversarial robustness in this work. During the rebuttal, based on one comment, we provided additional results on white-box PGD-20 adversarial robustness. The results in Table 2 represent the **best (highest) adversarial robustness**, which was only observed in the early stages of training. Considering the characteristics of LM-VP, in the early stages of training, PGD-20 has not fully utilized the information of the victim model, and those values reflect more about the inherent robustness of different fixed pretrained models, thus showing no consistent pattern. However, **as training progresses, for all pretrained models, the adversarial robustness continues to decline until it reaches a stabilized status**. To support this, we provide the changing trend of adversarial robustness for 10 epochs, as supplemental results to Table 2 (see the table below).
> > >
> > > We can also refer to [1] and [2] for how pretrained models impact downstream adversarial robustness. However, their research is limited to ResNet50 or Swin, and Yamada et.al conclude that **“network architecture is a strong source of robustness when we consider transfer learning”** [2]. If different pretrained models yield different results, it becomes challenging to draw a consistent conclusion on the adversarial robustness of LM-VP, making it even more difficult to study its trade-off with privacy.
> > >
> > > **Response to the final points of the reviewer**:
> > >
> > > 1. I understand and agree with this statement. Our rebuttal Table 2 is based on PGD-20 attacks but only includes the best adversarial robustness. From the table below, after certain iterations, adversarial robustness tends to be stable and potentially reflects the worst-case performance for LM-VP models. In contrast, transfer adversarial robustness remains stable during training, the worst-case adversarial robustness is definitely lower than transferred adversarial robustness.
> > >
> > > 2. "If you find ... in your experimental setup." In our previous rebuttal Table 2, the best adversarial robustness result leads to the misunderstanding. Our additional results about **epoch-wise changes in adversarial robustness** of that experiment prove that white-box attacks are stronger than transfer attacks (see the table below).
> > >
> > > We agree with the statement that "transferred adversarial robustness cannot reflect the overall adversarial robustness." However, the focus of this paper is to study the trade-off between transfer adversarial robustness and privacy. We believe that transferred adversarial robustness is a very important attribute to study, especially **when different pretrained models are used**, where transferred adversarial attacks can provide more consistent evaluation results.
> > >
> > > In the paper, we state "a better trade-off between adversarial robustness and privacy." Here, adversarial robustness specifically refers to transfer adversarial robustness. Please refer to our response to Question 1,  we only consider transfer adversarial robustness in our manuscript. As mentioned, in the next version of the manuscript, we will **add "transferable" or "transferred" in places where it was omitted** in the current version.
> > >
> > > 3. As we clarified in our rebuttal, we did not consider white-box adversarial robustness in this work, since the main purpose of this paper is to explore the trade-off between transfer adversarial robustness and privacy for LM-VP models, especially where different pre-trained models are employed, and we demonstrate that transferred AT can improve both simultaneously.
> > >
> > > |Epoch|1|2|3|4|5|6|7|8|9|10|
> > > |:-:|:-:|:-:|:-:|:-:|:-:|:-:|:-:|:-:|:-:|:-:|
> > > |ResNet50|7.37|**8.33**|5.28|3.13|1.80|0.78|0.33|0.29|0.16|0.25|
> > > |ResNet152|**57.09**|56.12|50.49|42.13|37.09|30.12|22.83|20.97|19.81|20.77|
> > > |WideResNet|**40.29**|39.26|30.28|26.65|21.34|17.88|15.08|12.24|12.33|10.09|
> > > |ViT|**19.28**|17.09|13.09|9.88|6.70|5.27|5.00|3.28|1.90|2.01|
> > > |Swin|0|0|0|0|0|0|0|0|0|0|
> > >
> > > [1] Vaishnavi P, et.al. A Study of the Effects of Transfer Learning on Adversarial Robustness[J]. TMLR.
> > >
> > > [2] Yamada Y, Otani M. Does robustness on imagenet transfer to downstream tasks?[C]//Proceedings of the IEEE/CVF conference on computer vision and pattern recognition.

---

> > > > ### Comment · Reviewer_J3te · 2024-08-13
> > > > **Thank you for your clarification**
> > > >
> > > > Dear authors and AC,
> > > >
> > > > Thank you for your detailed response on transferred adversarial robustness and the specialty of the pretrained model effects on downstream tasks. I would like to raise my score to 4 pts.
> > > >
> > > > I still hope that in the future works, the discussion around white-box adversarial robustness can be included because the current version seems to mix up the transferred adversarial robustness and adversarial robustness. Like the title for this work, "Towards Evaluation of **Adversarial Robustness** and Privacy on Label Mapping Visual Prompting Models", the first impression for me is definitely that you use LM-VP to improve both adversarial robustness and privacy at the same time.
> > > >
> > > > I will see the opinion by the AC in the next discussion phase.

---

> > > > > ### Author Response · Authors · 2024-08-14
> > > > >
> > > > > Thank you for raising the score and for your comments.
> > > > >
> > > > > We will incorporate the analysis of white-box adversarial robustness during the rebuttal into our manuscript to enhance our contribution. This will include the impact of different pretrained models on white-box adversarial robustness and the analysis of the failure of standard adversarial training on the LM-VP model, which we believe will bring more insights into the security assessment of the LM-VP model. We will also revise the title to ensure clarity and avoid ambiguity.

---

### Official Review · Reviewer_Jw71 · 2024-07-13

**Soundness:** 3
**Presentation:** 2
**Contribution:** 3
**Rating:** 5
**Confidence:** 3

**Summary:**

Adversarial robustness and privacy are important considerations in AI security, particularly in deep learning models. Adversarial training (AT) is effective in enhancing robustness against attacks, but it increases vulnerability to membership inference attacks (MIAs), compromising privacy. This trade-off between robustness and privacy highlights the need for evaluation. Visual prompting, a model reprogramming technique, shows promise in vision tasks, but its performance under attacks and MIAs requires further assessment. This study evaluates the joint adversarial robustness and privacy of label-mapping-based visual prompting (LM-VP) models, combined with transferred AT, demonstrating a favorable trade-off between the two.

**Strengths:**

1. The article provides a comprehensive evaluation of the security aspects, specifically adversarial robustness and privacy, of Label Mapping Visual Prompting (LM-VP) models, contributing valuable insights to the field of deep learning security.
2. It introduces the concept of transferred adversarial training (AT) for LM-VP models, offering a novel approach to enhancing adversarial robustness while maintaining privacy, which can have significant implications for improving the security of deep learning models.

**Weaknesses:**

1. The article lacks theoretical support and interpretability for the analysis of LM-VP models, which may limit the depth of understanding of the security implications of these models.
2. The evaluation primarily relies on empirical findings, which may not fully capture the theoretical underpinnings of the observed relationships between adversarial robustness and privacy in LM-VP models.

**Questions:**

Can the authors provide more detailed insights into the theoretical foundations and assumptions underpinning the empirical findings on the trade-off between adversarial robustness and privacy in Label Mapping Visual Prompting (LM-VP) models?

**Limitations:**

The article does not extensively discuss the theoretical assumptions and proofs underlying the empirical findings, potentially limiting the generalizability and robustness of the conclusions drawn. While the paper addresses the trade-off between adversarial robustness and privacy in LM-VP models, it may not fully explore the broader societal impacts and ethical considerations of deploying such models in real-world applications.

---

> ### Author Rebuttal · Authors · 2024-08-07
>
> **W1 and W2**.  The main novelty of this work does not lie in the theoretical aspect. The main contribution of our work is actually to introduce a novel method to jointly improve the transfer adversarial robustness and privacy of LM-VP models. This issue has not been fully explored before and we are the first to study the robustness and privacy trade-off of the LM-VP model, in this paper, we introduce the transfer AT for LM-VP models and conduct comprehensive experiments varying different pre-trained models to valid the effectiveness of transfer AT compared with standard training on LM-VP models, i.e., jointly improve the transfer adversarial robustness and training data privacy.
>
> On the other hand, we admit that a theoretical understanding of the interaction between adversarial robustness and privacy for LM-VP models is a significant research problem. Yet, it is still an open challenge in the community. Song et.al [1] tried to analyze this interaction on general machine learning models. Through extensive experiments, they concluded that **larger generalization errors and larger training data sensitivity make the model more susceptible to MIA** but did not provide any principled theoretical analysis. From Table 1 and Table 3 in our manuscript, the generalization error on the train and test accuracy may not be a key factor that influences the training data privacy for LM-VP models, as each pre-trained model does not show a significant generalization error, but their MIA values vary.
>
> **Q1 and L1**. We think some possible insights can be explored to tackle the theoretical challenges, for example: (1) During transfer AT, the original training examples are perturbed before feeding into the model, which means these data are not exposed to the trained model, this may be one factor that transfer AT helps mitigate the MIA issue because LM-VP models do not suffer from large generalization error and increased training data sensitivity (Table1, 2 and Figure3 in our manuscript) and transfer AT do not train the original training examples.
>
> (2) Differential privacy (DP) is a method to guarantee privacy, both adversarial examples and VP introduce noise to original images, which may potentially resemble some operation of DP, thus mitigating the MIA issue. Other aspects, such as dataset complexity, training data size, pre-trained model architecture, and different adversarial training strategies will also be important considerations for future work to build the theory.
>
> [1] Liwei Song, Reza Shokri, and Prateek Mittal. Privacy risks of securing machine learning models against adversarial examples. In Proceedings of the 2019 ACM SIGSAC Conference on Computer and Communications Security, pages 241–257, 2019.

---

> > ### Comment · Reviewer_Jw71 · 2024-08-09
> > **About the novelity**
> >
> > Thanks for considering my concerns.
> >
> > However，the author doesn't seem to understand what I'm trying to say. I know, in this article, the main contribution is to evaluate the adversarial robustness and privacy of the LM-VP models, using the method of Transferable AT. I want to ask why Transferable AT can be used for evaluation the boundary of adversarial robustness and privacy, and how is it modeled and quantified? In my opinion, this paper only uses an existing method to evaluate a complex problem. Maybe, further discussions about the motivation and original contributions rather than more experiments should be highlighted.

---

> ### Author Response · Authors · 2024-08-12
>
> We sincerely thank the reviewer for the valuable feedback.
>
> The idea of this paper, i.e., studying the relation of the adversarial robustness and privacy within the LM-VP model stems from similar concerns observed in general trained models. In the context of general trained models, a key consideration is the boundary relationship between standard adversarial robustness and privacy. However, existing research [1] highlights a conflict between these boundaries when employing white-box adversarial attacks. As a result, in this paper, we use the transferred adversarial attacks to study the robustness instead of using white-box attacks, which from our point of view, is more sensible; the specific reasons are listed below:
>
> **1**. A crucial distinction between the LM-VP model and a general model lies in the presence of a pre-trained model that does not participate in training [2]. Evaluating the LM-VP model using white-box adversarial robustness metrics can be heavily influenced by the choice of pre-trained model, as shown in Table 2 of the PDF. There is no clear pattern in their best adversarial robustness, potentially leading to different boundary relationships between adversarial robustness and privacy. Moreover, since we train the LM-VP model on the target dataset, but the generation of adversarial examples relies on the fixed pre-trained model from the source dataset domain, as a result, different pre-trained models may result in varying boundary performances, i.e., in this sense, using white-box adversarial attacks for the evaluation would make it difficult to draw a consistent conclusion.
>
> **2**. In contrast, when considering the transferred adversarial robustness of LM-VP models, the intensity of the transfer attack remains constant once the attack model is selected. This consistency holds regardless of the chosen pre-trained model, ensuring that the transfer adversarial training process consistently optimizes in the same direction. This inherent consistency thus is more helpful for exploring and establishing a sensible boundary relationship between transferred adversarial robustness and privacy within LM-VP models. Therefore, utilizing transferred adversarial attacks serves as a more reliable and insightful evaluation method in this context.
>
> **Modeling Approach**:
> Within the framework of transfer AT, the LM-VP model, comprising VP (Visual Prompt), a pre-trained model, and LM (label mapping), is treated as a unified black-box system. A fixed-parameter attack model, excluded from the training process, is employed to conduct transferred adversarial attacks. Under this setup, this paper systematically evaluates both standard training and transfer AT across LM-VP models that are based on various pre-trained models, with a focus on their impact on the trade-off between transfer adversarial robustness and privacy.
>
> **Quantification and Analysis**:
> To quantify the boundary relationship between robustness and privacy, we leverage numerical metrics of transfer adversarial robustness and the success rate of membership inference attacks. By systematically comparing these metrics under both standard training and transfer AT across a range of pre-trained models, we aim to arrive at a consistent and generalizable conclusion. This analysis seeks to demonstrate that transfer AT effectively achieves a superior balance between transfer adversarial robustness and privacy boundaries, ultimately establishing it as a more secure training approach for LM-VP models.
>
> [1] Liwei Song, Reza Shokri, and Prateek Mittal. Privacy risks of securing machine learning models against adversarial examples. In Proceedings of the 2019 ACM SIGSAC Conference on Computer and Communications Security, pages 241–257, 2019.
>
> [2] Bahng H, Jahanian A, Sankaranarayanan S, et al. Exploring visual prompts for adapting large-scale models[J]. arXiv preprint arXiv:2203.17274, 2022.

---

> > ### Comment · Reviewer_Jw71 · 2024-08-13
> > **Good clarification**
> >
> > Authors provide a clear clarification for my concerns in this round, I hope you can add these rebuttal texts into the revised version if it is accepted.  Blackbox evaluation is indeed a bigger challenge than whitebox.

---

> > > ### Author Response · Authors · 2024-08-14
> > >
> > > Thank you for your suggestion and for raising the score. We will add the analysis of white-box adversarial robustness during the rebuttal into our revised manuscript.

---

### Author Rebuttal · Authors · 2024-08-07

We thank the reviewers for their valuable comments on our work. We are very grateful to the reviewers for their recognition of our research topic and for their suggestions to improve our work. We give specific responses to each of the reviewers' comments. If there are further questions, we are happy to communicate with the reviewers further.
In short, our main responses are as follows:
1. **Robustness-privacy trade-off in LM-VP models**: We analyze the differences between the LM-VP model and general models in the robustness and privacy trade-off and provide some insights to explain why transfer AT enables the LM-VP model to achieve joint improvements on transferred adversarial robustness and privacy. Please refer to our formal response to Reviewer Jw71.
2. **Additional datasets**: According to the suggestion of Reviewer J3te, we add the experimental results of the LM-VP model on TinyImageNet. The conclusion is consistent with that on CIFAR-10, results can be seen in Table 1 in the PDF.
3. **ConvNext and EVA performance**: Although ConvNext and EVA achieve high transfer adversarial robustness after transfer learning using LM-VP or fine-tuning, they are vulnerable to MIA, and transfer AT improves their MIA resistance. Please refer to our formal response to Reviewer J3te.
4. **White-box adversarial robustness of LM-VP models**: Based on the question raised by Reviewer J3te, we analyze the performance of LM-VP models under white-box adversarial attacks using standard training and standard AT. The experimental results are shown in Table 2 in the PDF. We further explain why standard AT is not suitable for the LM-VP model. Please refer to our formal response to Reviewer J3te. (We use standard AT here to distinguish the transfer AT in our manuscript)
5. **Ablation experiment on epsilon size 4/255**: According to the suggestion of Reviewer RUkH, we complete preliminary experiments on epsilon size 4/255 and will complete the experiments of other pre-trained models in our manuscript, results are shown in Table 3 in the PDF.

We provide the main experimental results during the rebuttal phase in the attached PDF file. These contents will be further improved and incorporated into the revised version of our manuscript.

---

> ### Comment · Area_Chair_hV4Y · 2024-08-11
>
> Dear Reviewers,
>
> The authors have responded to your valuable comments.
> Please take a look at them!
> Thanks!
>
> Best,
> AC

---

### Decision · Program_Chairs · 2024-09-25

**Decision:**

Accept (poster)

**Comment:**

In this manuscript, the authors aim to introduce a new method to address both the transfer adversarial robustness and privacy of LM-VP models simultaneously.
The motivation and contribution are clear.
As for the reviewers' concerns, including novelty, white-box adversarial robustness, more experiments, unclear wordings, etc, they have been properly clarified.
Overall, this manuscript is suggested to accepted.
However, in responding to the concerns of Reviewer J3te and other reviewers, the rebuttal contents must be included in the revised version.